# Effects of a Motor Imagery Task on Functional Brain Network Community Structure in Older Adults: Data from the Brain Networks and Mobility Function (B-NET) Study

**DOI:** 10.3390/brainsci11010118

**Published:** 2021-01-17

**Authors:** Blake R. Neyland, Christina E. Hugenschmidt, Robert G. Lyday, Jonathan H. Burdette, Laura D. Baker, W. Jack Rejeski, Michael E. Miller, Stephen B. Kritchevsky, Paul J. Laurienti

**Affiliations:** 1Sticht Center for Healthy Aging and Alzheimer’s Prevention, Department of Internal Medicine Section on Gerontology and Geriatric Medicine, Wake Forest School of Medicine, Winston-Salem, NC 27103, USA; chugensc@wakehealth.edu (C.E.H.); ldbaker@wakehealth.edu (L.D.B.); skritche@wakehealth.edu (S.B.K.); 2Department of Radiology, Wake Forest School of Medicine, Winston-Salem, NC 27103, USA; rlyday@wakehealth.edu (R.G.L.); jburdett@wakehealth.edu (J.H.B.); plaurien@wakehealth.edu (P.J.L.); 3Department of Health and Exercise Science, Wake Forest University, Winston-Salem, NC 27109, USA; rejeski@wfu.edu; 4Department of Biostatistics and Data Science, Wake Forest School of Medicine, Winston-Salem, NC 27103, USA; mmiller@wakehealth.edu

**Keywords:** functional brain imaging, networks, brain, modularity, mobility, aging, motor imagery, SPPB

## Abstract

Elucidating the neural correlates of mobility is critical given the increasing population of older adults and age-associated mobility disability. In the current study, we applied graph theory to cross-sectional data to characterize functional brain networks generated from functional magnetic resonance imaging data both at rest and during a motor imagery (MI) task. Our MI task is derived from the Mobility Assessment Tool–short form (MAT-sf), which predicts performance on a 400 m walk, and the Short Physical Performance Battery (SPPB). Participants (*n* = 157) were from the Brain Networks and Mobility (B-NET) Study (mean age = 76.1 ± 4.3; % female = 55.4; % African American = 8.3; mean years of education = 15.7 ± 2.5). We used community structure analyses to partition functional brain networks into communities, or subnetworks, of highly interconnected regions. Global brain network community structure decreased during the MI task when compared to the resting state. We also examined the community structure of the default mode network (DMN), sensorimotor network (SMN), and the dorsal attention network (DAN) across the study population. The DMN and SMN exhibited a task-driven decline in consistency across the group when comparing the MI task to the resting state. The DAN, however, displayed an increase in consistency during the MI task. To our knowledge, this is the first study to use graph theory and network community structure to characterize the effects of a MI task, such as the MAT-sf, on overall brain network organization in older adults.

## 1. Introduction

Maintaining mobility with aging is important to both quality of life and independence, yet numerous questions remain about the causes of unexpected decline [1]. This gap in knowledge becomes increasingly pressing given the rapidly growing aging population [2]. Much of the work aimed at understanding the development of mobility disability has focused on balance, cardiorespiratory fitness, or muscle strength. It has been suggested that investigating changes in the contributions of the brain and central nervous system to mobility in older adults will provide an important new understanding of how mobility disability arises and how it relates to aging-related cognitive decline [3]. In particular, changes in functional brain networks have gained increasing attention as they may precede irreversible structural changes and have been associated with mobility in older adults [4,5].

Intrinsically correlated motor networks derived from functional magnetic resonance imaging (fMRI) can be detected at rest, and properties of motor networks at rest are associated with measures of mobility outside the scanner [4,5]. Networks identified during rest are often highly replicable, which provides a reliable tool to link complex brain connectivity patterns with human behaviors [6]. However, they are inherently limited in their ability to reflect the dynamic network reconfiguration occurring during a task [7,8]. Task-based fMRI has been widely adopted to characterize the activity of brain regions involved with overt movement. However, task-based fMRI that includes movement (e.g., finger-tapping) is susceptible to a variety of challenges such as motion artifacts, which degrade image quality.

Motor imagery (MI) tasks, or the imagined movement of the body while the muscles are not engaged, have been demonstrated to approximate the activation in the brain during overt movement [9,10,11]. A variety of MI tasks have been reported in the literature. Despite being conducted mentally, imagined movement has been shown to initiate activation in brain regions that are engaged during actual movement [12,13]. This includes increased activation in regions such as the supplementary motor area (SMA), premotor cortex (PMC), dorsolateral prefrontal cortex (DLPFC), basal ganglia (BG), and parietal cortex [14,15,16,17,18]. The primary motor cortex (M1) is reported to be more variably activated during MI tasks than during motor execution [9,18]. Although these efforts have advanced our understanding of brain activation during a MI task, few studies have considered task-related changes during MI from a network perspective. Given that mobility requires the precise coordination of multiple brain regions, network studies are necessary to untangle these complex interactions [19,20]. Understanding not only which brain regions become active but also how they interact could be foundational to elucidating the neural correlates of mobility.

Here, we advance the existing literature on neural processes associated with MI using an adaptation of the Mobility Assessment Tool–short form (MAT-sf) to examine brain networks associated with MI. The MAT-sf is a brief, self-report tool with 10 animated video clips illustrating common mobility tasks [21,22,23]. It is correlated with major objective tests of physical function and has been found to predict outcomes following surgery and major mobility disability [16,22,23,24]. In the present study, we utilize a variation of the MAT-sf adapted for use in the scanner in which two videos are shown to participants that include relatively easy and more difficult motor tasks, respectively. Given the established relationships between the MAT-sf and real-world mobility, this allows for a more ecologically valid study of brain network architecture during overt movement.

In an attempt to address the gaps in the MI literature, we utilize graph theory methods in a well-characterized sample of over 150 community-dwelling older adults in an ongoing observational study on the neural correlates of mobility in aging. Our analyses are cross-sectional and focus on both global and regional network community structure differences between resting state and two difficulty levels of the MI task. Given our prior work on the community structure of the sensorimotor network (SMN), we hypothesized that the consistency of the SMN will increase during the MI task. Considering the widespread effect of MI tasks in the brain, it is unlikely that changes in network consistency will be solely localized to the SMN. Of note, the default mode network (DMN) and dorsal attention network (DAN) are possible candidates for significant modification during the MI task. The DMN is known to show reductions in activity when transitioning from the resting state to a task [25]. Conversely, the DAN is known to become active during tasks that require visuospatial attention [26]. Consequently, we further hypothesized that the consistency of the DMN will decrease during the MI task, whereas the consistency of the DAN will increase due to their established dynamics. This is, to the best of our knowledge, one of the first brain network studies on MI tasks, and so our aims were to characterize these global and regional patterns of brain connectivity during the task and compare them with existing literature.

## 2. Materials and Methods

### 2.1. B-NET Study Design

Data were from the Brain Networks and Mobility (B-NET) study (NCT03430427). B-NET is an ongoing longitudinal, observational trial of community-dwelling older adults aged 70 and older recruited from Forsyth County, NC and the surrounding regions. The participants were asked to complete a baseline study visit along with follow-up visits at 6, 18, and 30 months. Brain MRIs were collected at baseline and the 30-month follow-up visit. Baseline cross-sectional data are presented.

### 2.2. Participants

B-NET recruitment is ongoing. The analysis presented here includes 157 participants with complete imaging data at baseline. To be included in the study, the participants had to be above the age of 70, willing to sign an informed consent form, and able to communicate with study personnel. Exclusion criteria included being unwilling or unable to have an MRI brain scan, being a single or double amputee, having musculoskeletal implants severe enough to impede functional testing (e.g., joint replacements), or dependency on a walker or another person to ambulate. The participants were also excluded if they had undergone surgery or hospitalization within the past 6 months; serious or uncontrolled chronic disease (stage 3 or 4 cancer, stage 3 or 4 heart failure, liver failure or cirrhosis of the liver, uncontrolled angina, respiratory disease requiring the use of oxygen, renal failure requiring dialysis, diagnosis of schizophrenia, bipolar, or other psychotic disorders, or alcoholism (>21 drink per week)); clinical manifestation of a neurologic disease affecting mobility; or prior traumatic brain injury with residual deficits. In addition, potential participants with a history of brain tumors, seizures within the last year, major uncorrected hearing or vision problems, plans to relocate within the next two years, participation in a behavioral intervention trial, or evidence of impaired cognitive function were excluded. Cognitive impairment was defined based on scores on the Montreal Cognitive Assessment (MoCA). Briefly, scores of 26–30 on the MoCA were considered immediately eligible for enrollment. MoCA scores from 21 to 25 were reviewed by the study neuropsychologist to determine eligibility. Finally, scores of 20 or lower on the MoCA were considered ineligible to enroll in the study. The study was in compliance with the Declaration of Helsinki and written informed consent was obtained from each participant prior to any data collection. The study was approved by the Institutional Review Board (IRB) of the Wake Forest School of Medicine (IRB Protocol No.: IRB00046460; approval date: 27 August 2020).

### 2.3. Motor Imagery Experimental Design

The study participants lay supine in the MRI scanner with visual stimuli projected onto a screen viewed through a mirror. Stimulus presentation consisted of videos from the Mobility Assessment Tool–short form (MAT-sf) that were played through a standard media player. Additional details on the MAT-sf can be found in the Appendix A, and additional details of the fMRI task are below.

### 2.4. Stimuli

Two videos were created using the MAT-sf stimuli, one showing mobility tasks that are relatively easy (Easy Task) and the other showing more difficult mobility tasks (Hard Task). The Easy Task items included (1) slowly walking on a short, level course indoors; (2) slow walking on level ground outdoors; (3) walking up a short, low-grade incline using a handrail; and (4) walking up a set of 3 stairs using a handrail. The Hard Task items included (1) walking on a short, level indoor course at a fast pace; (2) walking up a flight of 10 stairs without using a handrail; (3) walking up a short, moderately inclined gravel slope outdoors; (4) walking down a flight of 10 stairs without using a handrail, and (5) slow jogging on a short, indoor course. Each task was administered for 260 s, where each stimulus was repeated 5 times.

### 2.5. Practice Sessions

Prior to entering the scanner, the participants received instructions and completed a practice session. A video of the MAT-sf MI task was played on a computer screen. The participants were oriented to the avatar, instructed to imagine themselves as the avatar, and told that their active involvement and engagement in the task is critical to the validity of the experiment. Once the participants understood task expectations, the study staff played example videos for the two task types. Each practice video lasted approximately one minute, and each item (movement displayed by the avatar) was demonstrated twice. Following each video, the participants completed a short survey with three questions on their ability to imagine themselves completing the task, which included: “How well were you able to imagine yourself doing the actual task?”, “Were you able to stay with the task the entire 4 min?”, and “The activities you just visualized have a similar level of difficulty. In general, could you perform tasks of that difficulty?”. Answers were collected on the visual analog scale (VAS), which included a slider with values ranging between 0 and 100 (0 = negative response, 100 = positive response).

### 2.6. Study Measurement Procedure

Immediately before entering the scanner, the study staff repeated the instructions from the practice session. Each MRI session began with several anatomical image acquisitions and a resting-state scan with a fixation on a cross. The order of the MI task runs was randomized to eliminate any potential biases [27]. Immediately before beginning the MI task scans, the study staff repeated the instructions from the practice session. The three self-report questions from the practice trial were asked after each MI scan, except the subjects were now asked if they were able to stay with the task the entire 4 min. The VAS was displayed on the screen viewed through the mirror and included a sliding marker that could be moved from 0 to 100 to illustrate the level of agreement with each question. The slider always began positioned at 50, and the study staff inquired if the participants would like to move the slider (left toward 0 or right toward 100). The participant verbally instructed the study staff through a microphone to increase or decrease the VAS indicator displayed on the screen. At the beginning of the study (the first 113 participants), the VAS responses were collected using the AVAS software [28]. After a software update on the stimulus delivery computer, we encountered difficulties with the AVAS software and switched to the REDCap software [29] to display the VAS and record responses. The experimental design is illustrated in Figure 1.

### 2.7. Magnetic Resonance Imaging

All brain images were collected on a Siemens 3T Skyra MRI Scanner equipped with a 32-channel head coil. Each scan session lasted approximately 1 h. Structural scans included a T1-weighted anatomical image (TR = 2300 msec; TE = 2.98 msec; number of slices = 192; slice thickness = 1.0 mm; voxel dimensions = 1.0 × 1.0 × 1.0 mm; FOV = 256 mm; scan duration = 312 s) obtained using 3D volumetric MPRAGE. Blood oxygenation level-dependent (BOLD) imaging (TR = 2000 ms; TE = 25 ms; number of slices = 35; slice thickness = 5.0 mm; voxel dimensions = 4.0 × 4.0 × 5.0 mm; FOV = 256 mm) was collected at rest (scan duration = 460 s) and during two motor imagery tasks (260 s each). BOLD Scans were collected parallel to the anterior commissure–posterior commissure (AC-PC) using multislice gradient-echo planar imaging (EPI) [30].

### 2.8. fMRI Preprocessing

Structural image segmentation, including the removal of nonbrain tissue and cerebrospinal fluid (CSF), was completed using Statistical Parametric Mapping version 12 (SPM12, http://www.fil.ion.ucl.ac.uk/spm). A whole-brain mask was created for each subject, using the white and gray matter segments, to improve the coregistration of the functional and anatomical images. All images were inspected and manually cleaned to remove any extraparenchymal tissues using the MRIcron software (https://www.nitrc.org/projects/mricron). Two observers manually checked the mask to ensure accurate full-brain coverage. High-resolution T1-weighted images were spatially normalized to the Montreal Neurological Institute (MNI) template using Advanced Normalization Tools (ANTs) [31]. Distortion correction was completed using FMRIB’s Software Library (FSL, www.fmrib.ox.ac.uk/fsl). The first 10 volumes of the BOLD images were dropped to allow for signal normalization. Slice time correction and realignment of the functional images were completed using SPM12. The BOLD images were coregistered to the native-space anatomical images and warped to MNI space using the transformation derived from ANTs. Head motion was additionally corrected for with motion scrubbing [32]. Finally, data were band-pass-filtered (0.009–0.08 Hz) to account for low-frequency drift and physiologic noise. Confounding signals were regressed out from the filtered data, including signals from white matter, gray matter, CSF, and 6 rigid-body motion parameters generated during the realignment process.

### 2.9. Calculation of Brain Networks

The presence of a connection between two nodes (i and j) was determined by completing a time series regression analysis. This produced a cross-correlation matrix, which represents the Pearson’s correlation coefficient that describes the connectivity between every network node. A threshold was then chosen to dichotomize the data and create the final binary adjacency matrix (Aij) where values above the threshold indicate a connection is present (e.g., correlation coefficients are transformed to either a 0 indicating no connection or a 1 where there is a connection between nodes). Aij is a n × n matrix where n is the number of brain voxels (~20,000). This matrix notes the presence or absence of a connection between any two nodes (represented by i and j). The threshold ensures that the density of connections was comparable across all the participants. More specifically, S = log (N)/log (K) is conserved across the participants. Here, N represents the number of network nodes, and K is the average node degree. Our networks were thresholded at S = 2.5 based on previous work showing that this threshold produces networks with density ratios that most closely compare to other naturally generated networks [33].

### 2.10. Brain Network Analysis

Community structure analyses were performed on each participant’s brain network to assign each voxel to a network community. A community is defined as a group of nodes that have more connections with each other than to other nodes in the total network [4]. The strength of community structures was identified using network modularity or Q [34]. A dynamic Markov process [35] was used to maximize Q and partition the brain networks into communities. As there is a stochastic component to the modularity algorithm, it was run 100 times, and the partition associated with the highest Q value was chosen for that given study participant. Following partitioning an individual’s brain network, each voxel was assigned the appropriate community membership and the spatial maps for the community organization were used in the second-level group analyses as described below.

Scaled Inclusivity (SI), a statistic utilized to compute the spatial consistency of community structure across a group of participants [36], was used for group analyses. Values of SI range between 0 and 1, and perfect spatial alignment of a community across all the participants would result in a value of 1 for all nodes in that community. Practice communities are not made of exactly the same nodes across the study participants as the spatial patterns vary across subjects due to differences in the community organization. Thus, in practice, each node is assigned a value less than 1 depending on the discordance across subjects (for a detailed description of this method applied to the brain see [37]). High SI values indicate that a particular network community is stable across the group and occupies comparable brain regions in each person. SI values can be compared between groups or conditions to identify population differences in community organization.

Here, we performed SI analyses using regions of interest (ROIs) to define a priori intrinsic brain subnetworks. ROIs were generated using resting-state brain network data in 22 normal young adults from a prior study [38]. We selected intrinsic networks that encompassed the entire brain and are commonly identified in functional atlases [39], including: the three subnetworks of a priori interest dorsal attention network (DAN), default mode network (DMN), sensorimotor network (SMN), as well as the central executive network, basal ganglia network, salience network, and visual network [38]. In addition to these 7 intrinsic networks, a region of interest (ROI) was generated that encompassed image regions with high MRI artifact incidence across the participants. Together, the 8 ROIs included all voxels in the brain. To address potential bias introduced with the use of ROIs generated in young adults, we generated study-specific ROIs utilizing a method conceptually similar to that commonly used in voxel-based morphometry to generate study-specific templates (VBM) [40]. First, SI was computed across all scan conditions (Rest, Easy, Hard) and all the participants for each of the 8 ROIs. This process generated 8 SI maps, one associated with each ROI. The SI value in each voxel was compared across the 8 SI maps and each voxel was assigned to the ROI with the highest SI value. One benefit of this method is that because the SI method uses the ROI as a seed but calculates overlap values across the entire brain for that ROI, it is possible to detect consistent differences in community structure that may be present in the subjects but not the original ROI. This process generated 8 study-specific ROI maps with each voxel belonging to only one ROI. The ROI encompassing high artifact regions was excluded from all further analyses.

Analyses were then performed to compare global and regional SI across the three study task conditions using the study-specific ROIs. First, SI was computed for each ROI and each task condition in each study participant. Global SI was computed for each condition by summing the SI values across all ROIs in each image voxel and then computing the whole-brain mean. Global values were statistically compared across the three conditions. Regional analyses were performed to assess the spatial organization of the following three intrinsic networks: the sensorimotor network (SMN), the default mode network (DMN), and the dorsal attention network (DAN). The study-specific ROIs for the SMN, DMN, and DAN can be found in the Appendix A. The 3 networks were chosen for the regional analyses based on a priori hypotheses. The SMN was chosen due to its established involvement in motor function and our prior work [4]. The DMN was chosen as it is known to exhibit connectivity and activity changes when going from rest to task [25]. The DAN was chosen because the MI task required visuospatial attention [26]. In order to control for the effect of the global change in SI and identify regional changes between task conditions, the SI maps were scaled prior to performing statistical comparisons. Scaling was performed independently for each condition and participant using min-max normalization across all SI maps. Once scaled maps were generated for each participant, the effect of the task condition on the spatial distribution of each of the three subnetworks was assessed using a permutation statistic [41].

### 2.11. Statistical Analyses

Differences in whole-brain and ROI network community structure consistency were statistically compared between scan conditions using Proc Mixed of SAS version 9.4 (SAS Institute Inc., Cary, NC, USA). A linear model for repeated measures was used to estimate differences between fMRI task conditions in global and regional consistency with age, sex, and race as covariates, and an unstructured covariance matrix was used to account for the covariance between repeated measures. Least-squares means, accounting for age, sex, and race, and the differences between these means, are provided. Bonferroni-adjusted *p*-values and confidence intervals are used to account for pairwise comparisons applied to global and regional consistency measures. β is used to represent differences in least-squares means.

## 3. Results

### 3.1. Baseline Characteristics

One-hundred-and-fifty-seven participants (mean age = 76.1 ± 4.3; % female = 55.4; % African American = 8.3; mean education = 15.7 ± 2.5) participated in the present study. Additional participant characteristics are illustrated in Table 1.

### 3.2. Global Community Structure Assessment

Analyses first focused on differences in average whole-brain community structure (assessed using global SI) between the three fMRI task conditions and were adjusted for age, sex, and race. Figure 2 illustrates the decline in global community structure across the three scan conditions. Overall, there is a general decrease in network consistency with the addition of the motor imagery task and with increased difficulty. Network consistency was lower in Hard Task than Rest ([β = 0.0118, 95% CI: [0.0072, 0.0163], *p* < 0.0001]) or the Easy Task ([β = 0.0079, 95% CI: [0.0034, 0.0124], *p* < 0.0001]). The difference between network consistency at Rest and the Easy Task did not reach statistical significance ([β = 0.0039, 95% CI: [−0.0008, 0.0085], *p* = 0.1356]). Results of the linear models estimating differences in the SI statistic can be found in Table 2.

### 3.3. Regional Community Structure Assessment

We next performed analyses to evaluate the network community structure within the DAN, DMN, and SMN networks across tasks adjusting for age, race, and sex. Figure 3 illustrates the change in network community structure consistency between scan conditions for the DAN, DMN, and SMN ROIs. Output from the linear models estimating differences in the SI statistic between scan conditions can be found in Table 2.

DMN was most consistent at Rest with significant declines observed during the Easy Task (Rest = 0.4021, Easy = 0.2202, [β = 0.1819, 95% CI: [0.1406, 0.2231], *p* < 0.0001]) and the Hard Task (Rest = 0.4021, Hard = 0.2096, [β = 0.1925, 95% CI: [0.1493, 0.2357], *p* < 0.0001]). There was no significant difference in DMN consistency between the Easy and Hard Tasks (Easy = 0.2202, Hard = 0.2096, [β = 0.0106, 95% CI: [−0.0196, 0.0408], *p* = 1.0]).

Consistency of the DAN was highest during the Easy Task, where consistency across subjects was greater than in Rest (Rest = 0.3159, Easy = 0.4047, [β = −0.0889, 95% CI: [−0.1231, −0.0545], *p* < 0.0001]) or the Hard Task (Easy = 0.4047, Hard = 0.3172, [β = 0.0875, 95% CI: [0.0503, 0.1247], *p* < 0.0001]). No significant difference was observed between Rest and the Hard Task (Rest = 0.3159, Hard = 0.3172, [β = −0.0013, 95% CI: [−0.0384, 0.0357], *p* = 1.0]).

SMN consistency was highest at Rest and significantly higher than in the Easy Task (Rest = 0.2862, Easy = 0.2369, [β = 0.0493, 95% CI: [0.0193, 0.0794], *p* = 0.0003]) or the Hard Task (Rest = 0.2862, Hard = 0.2483, [β = 0.0379, 95% CI: [0.0057, 0.0700], *p* = 0.0150). No significant difference was observed between SMN consistency when comparing the Easy and Hard conditions (Easy = 0.2369, Hard = 0.2483, [β = −0.0115, 95% CI: [−0.0353, 0.0123], *p* = 0.7353).

### 3.4. VAS Reporting

To compare self-reported task performance scores between task conditions, for each of the three self-report questions, we subtracted the VAS ratings of the Hard Task from the Easy Task for each participant. We then grouped the results based on if they were negative (Easy Task score < Hard Task score), zero (Easy Task score = Hard Task score), or positive (Easy Task score > Hard Task score) (Table 3). Briefly, when asked if they were able to imagine themselves completing the tasks in the videos, 29.3% of the participants reported more difficulty imaging in the Hard Task than the Easy Task. Compared to the Easy Task, 20.7% of the participants reported a lower ability to stay with the Hard Task for the full 4 min. Finally, when questioned about their ability to perform tasks of the difficulty shown in the videos, 36.7% of the participants reported a lower ability to perform the actions in the Hard Task than the Easy Task.

## 4. Discussion

In the present study, we investigated the effect of a MI task on the overall brain network using graph theory network analyses. The MI tasks employed items from the MAT-sf, a motor imagery task where performance is associated with major objective tests of mobility and predicts clinical outcomes, including major mobility disability [21,22,23,24]. We found that, when compared to the resting state, global community structure consistency declined during the MI tasks and with increased difficulty. In addition to the global decrease, regional variations in the consistency of subnetworks important for mobility were observed. The consistency of the DMN and SMN were highest during the resting state and showed significant declines with the addition of the MI task. The DAN, however, showed a significant increase in consistency with the addition of the task. Our results, showing a global decrease in network consistency following the onset of the MI task, likely reflect the complex coordination of brain regions required during movement and the complex movements included in the continuous MI task.

Our initial hypothesis was that the SMN would show increased network consistency during the MI task given prior literature showing the increased activation of motor-related regions during motor imagery [13,14,15]. Results from our prior work also demonstrated higher resting-state consistency of the SMN is associated with higher scores on the SPPB [4]. However, we observed significant declines in SMN consistency during the MI task. During our MI task, a collection of varying movements is executed that include both upper- and lower-body movements, each of which may possibly elicit activation in a distinct set of brain regions. Given that network consistency assesses the overlap of network modules, the administration of an MI task of complex movements could be expected to increase the variability in spatial patterns of network connectivity within somatomotor regions, leading to reduced SMN consistency when compared to the resting state.

The observation that the DMN had the highest consistency at rest with significant declines during the MI task confirmed our secondary hypothesis about the DMN, which was based on the established task-related dynamics of the DMN. The DMN was originally identified as a collection of regions that shows coordinated decreases in activity during the performance of goal-directed and attention-demanding tasks, such as the MAT-sf. [25]. In contrast to the DMN, the DAN increased consistency during the MI task, as was hypothesized. There is a general consensus that the DAN directs visuospatial attention and short-term memory processes [42]. Recent evidence has also pointed toward the role of the DAN in spatial orienting during a task [43]. Consequently, increased consistency of the network during a MI task, such as the MAT-sf, was anticipated.

Prior studies have proposed that the activation and deactivation of functional regions of the brain is a dynamic process that reveals networks that are anticorrelated [44,45,46,47,48,49,50]. This anticorrelation is generally taken as an indication of competing functions, namely the DMN (“task-negative”) and DAN (included in “task-positive”), which showed a particularly strong anticorrelation [45,46,47,49]. Our results showed increased DAN consistency and decreased DMN consistency during the MI task. The findings within the DMN are in line with current theories of modularity in aging. With increased age, connectivity within the DMN is observed to decrease, which is associated with age-related cognitive decline [51]. These age-related decreases in connectivity within the DMN could translate to the reductions in network consistency observed in our findings. Anticorrelations between brain networks have also been characterized in studies on the relationships between functional connectivity and overt mobility. Decreased gait variability is associated with increased negative functional connectivity (anticorrelation) between the DMN and DAN [5]. This was interpreted to mean that an increased ability to sustain attention, as assessed by an increased anticorrelation between the two networks, improves an individual’s ability to maintain a steady gait. A second study published similar results with greater functional connectivity between the DMN and the frontoparietal network (FPN) [52]. The FPN shares characteristics with the DAN has been found to be associated with reduced performance on a finger-tapping task, likely due to increased distraction [52]. Overall, the interaction between the DMN, DAN, and other networks is complex and requires further study, especially in the context of mobility.

There was no significant difference in consistency of the DMN or SMN between the Easy Task and Hard Task conditions. This could indicate that the difficulty of the task being imagined does not play a notable role in the connectivity of these networks. The DAN, however, showed a significant decrease in consistency between the Easy and Hard Task after significantly increasing consistency between Rest and the Easy Task. This decrease may seem unexpected given that regression analyses of activation patterns show that the DAN increases activity with increasing attentional demands [26]. However, in these MI tasks, task difficulty refers to the physical demand required during the imagined movements and may not have the required additional attentional or cognitive resources to imagine. It is also possible that imagining tasks that might be at or near the limit of an individual’s physiological capacities elicits interactions between the DAN and additional attention-related networks. For example, there is evidence of interacting activity between the DAN and networks such as the ventral attention network (VAN) during visuospatial tasks [26,53,54]. DAN–VAN interactions during the Hard Task could potentially account for decreased consistency within the DAN.

Other potential explanations for decreased consistency of the DAN in the Hard Task compared to the Easy Task may relate to the ability of subjects to complete the imagined movements. Thirty-seven percent of the subjects indicated that they did not think they could complete the movements shown in the Hard Task. To test whether subjects were less able to imagine movements they could not complete, we assessed correlations between global, SMN, DMN, and DAN consistency and self-rated ability to complete the imagined tasks and found no meaningful or statistically significant associations. We also assessed whether self-rated ability to imagine the task correlated with global, SMN, DMN, or DAN consistency and, again, found no associations. It is possible that the self-report questions did not capture essential information or that there is not a simple linear relationship between these scores and network community structure. It is also possible that a change in emotional state during the task may be responsible for the decrease in consistency between the Easy and Hard conditions. Alterations in an individual’s emotional state can lead to widespread shifts in functional connectivity, including in the DAN and VAN [55], and positive emotional states are considered protective against age-related mobility impairments [56]. Variability in the emotional response to the Hard Task could therefore result in reduced DAN consistency across subjects. Assessment of emotional responses to mobility tasks through self-report or collection of physiological variables such as heart rate or galvanic skin response would allow a better understanding of the role of emotions in mobility function.

Our study benefited from a few key strengths. Firstly, the use of the MAT-sf provides a unique opportunity to study functional connectivity in the brain during a task that is validated as a predictor of major mobility disability and surgical outcomes, scores of the 400 m walk, and scores on the SPPB [21,22,23]. To our knowledge, we are the first to publish functional connectivity data using a task with high ecological validity such as the MAT-sf. Secondly, the collection of both resting-state and task-based data allowed for a direct comparison of the functional architecture between different states. Thirdly, the use of graph theory methods allowed for the assessment of intrinsic connectivity of motor regions during a continuous task and at rest, adding to our understanding of motor-related networks, which, so far, has been largely developed by examining task-related activations using linear regression methods. Finally, B-NET currently has over 150 participants and will collect follow-up scans at the 30-month visit. The sample size and longitudinal design will allow the opportunity to advance understanding of how mobility-related functional brain networks are associated with current mobility function and also change over time [57].

The current analysis also has several limitations. First, the data presented are cross-sectional. However, B-NET is designed for a follow-up scan during the 30-month follow-up, which will allow for longitudinal analyses. Second, we did not collect a nonmotor imagery task as a comparison for network connectivity during the MAT-sf. This would have allowed for disambiguation of activity specifically related to the motor components of the task from those specific to imagining. Lastly, there are limitations associated with the use of self-report to assess task performance. Subjects were asked to verbally guide the study staff in assessing their ability to imagine, perform the task, and attend for the full task duration using a VAS. However, it is possible that the questions selected did not collect all the relevant information to assess task engagement, that requiring a verbal response recorded by staff was not ideal, and the lack of trial-by-trial feedback limits the ability to assess any variations within the task over time. Efforts to mitigate these limitations included the study staff practicing the task with subjects before entering the scanner and providing explicit instruction to remain engaged and attentive during the task at three time points.

These findings provide opportunities for future studies designed to investigate interventions to improve mobility, which are critical to further elucidating the relationships between the brain and physical activity [58,59,60]. We have shown that the MAT-sf can be used as a MI task and evokes distinguishable patterns of connectivity from the resting state. When combined with the known correlations between the MAT-sf and scores on the 400 m walk and SPPB, this could provide an opportunity to study both cross-sectional and longitudinal questions on mobility disability during task visualization. Our protocol is also highly replicable and would thus be suitable for multisite studies focused on mobility. Additionally, the MAT-sf can be completed both inside and outside of the scanner, which could allow for continuous data collection relevant to the task between study visits in which brain imaging occurs.

## 5. Conclusions

To our knowledge, this is the first study to use graph theory and network community structure analyses to characterize the effects of a MI task that incorporates complex, everyday functional movements on brain network connectivity. Historically, research techniques have been limited in their ability to measure the neural mechanisms of movement due to technological and paradigm limitations. Our results highlight a network analysis and MI task that overcomes these shortcomings. Interpretation of these results should keep in mind the exploratory nature of the analyses. It is important to replicate these findings in other large, well-characterized cohorts. Overall, these findings provide fascinating insights into the neural correlates of mobility and highlight the complex interactions in the brain required during movement.

## Figures and Tables

**Figure 1 brainsci-11-00118-f001:**
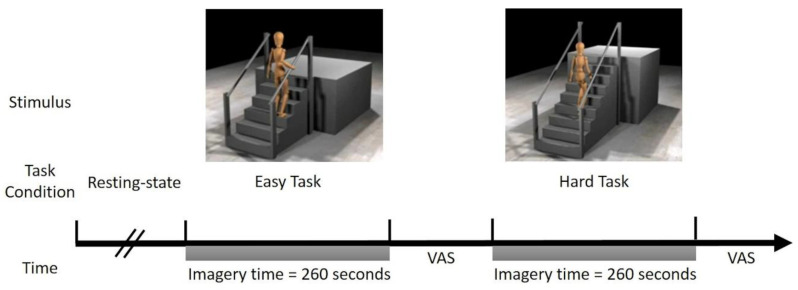
A single resting-state run was completed prior to the start of the motor imagery (MI) task. The order of the tasks was randomized across the participants; half the participants completed the Easy Task first and the other half the Hard Task. Participants were instructed to visualize themselves as the avatar in the videos as it completed a series of movements. Each task was played for 260 s. After each task, the participants completed self-report questions using a visual analog scale (VAS).

**Figure 2 brainsci-11-00118-f002:**
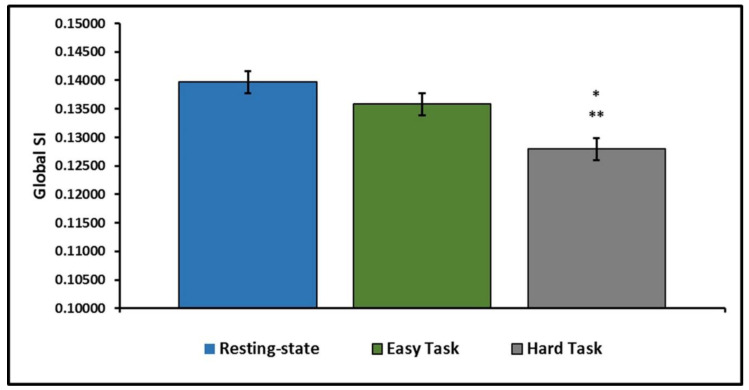
Network consistency is highest during resting-state and declines with the addition of the motor imagery task. Network consistency is significantly lower in the Hard Task when compared to both the Resting state and the Easy Task. * indicates a significant difference from the Resting state. ** indicates a significant difference from the Easy Task.

**Figure 3 brainsci-11-00118-f003:**
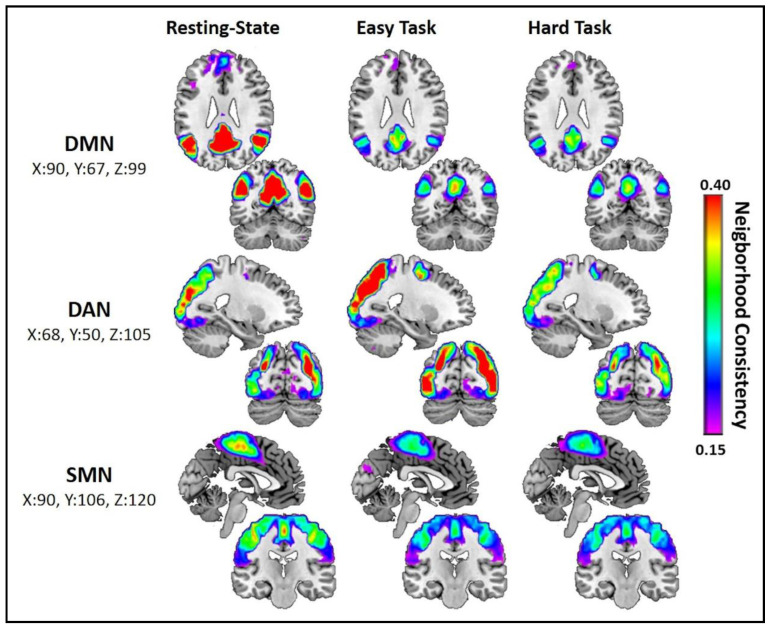
Cool colors indicate a less consistent network community structure than warm colors. The dorsal attention network (DAN) is more consistent during the MI tasks, particularly the Easy Task, than the Resting state. The default mode network (DMN) is most consistent during the Resting state and shows a sharp decline during the MI tasks. The sensorimotor network (SMN) is also more consistent during the Resting state than during MI tasks.

**Table 1 brainsci-11-00118-t001:** General characteristics of sample.

	Sample
*n*	157
Age (years ± STD)	76.1 ± 4.3
Women *n* (%)	87 (55.4)
African American n (%)	13 (8.3)
Education (years ± STD)	15.7 ± 2.5
SPPB Score	10.4 ± 1.6
MAT-sf	63.2 ± 9.2
BMI (kg/m^2^)	27.9 ± 5.4
MoCA (mean ± STD)	25.6 ± 2.2
DSST (mean ± STD)	56.1 ± 12.2

Note: SPPB = Short Physical Performance Battery; MAT-sf = Mobility Assessment Tool–short form; BMI = body mass index; MoCA = Montreal Cognitive Assessment; DSST = Digit Symbol Substitution Test.

**Table 2 brainsci-11-00118-t002:** Network community structure consistency comparison between task conditions.

Comparison	β	95% CI	*p*-Value
Global			
Rest vs. Easy	0.0039	[−0.0008, 0.0085]	0.1356
Rest vs. Hard	0.0118	[0.0072, 0.0163]	<0.0001
Easy vs. Hard	0.0079	[0.0034, 0.0124]	0.0001
DMN			
Rest vs. Easy	0.1819	[0.1406, 0.2231]	<0.0001
Rest vs. Hard	0.1925	[0.1493, 0.2357]	<0.0001
Easy vs. Hard	0.0106	[−0.0196, 0.0408]	1.0
DAN			
Rest vs. Easy	−0.0889	[−0.1231, −0.0545]	<0.0001
Rest vs. Hard	−0.0013	[−0.0384, 0.0357]	1.0
Easy vs. Hard	0.0875	[0.0503, 0.1247]	<0.0001
SMN			
Rest vs. Easy	0.0493	[0.0193, 0.0794]	0.0003
Rest vs. Hard	0.0379	[0.0057, 0.0700]	0.0150
Easy vs. Hard	−0.0115	[−0.0353, 0.0123]	0.7353

**Table 3 brainsci-11-00118-t003:** Comparison of VAS scores between the Easy and Hard MI Tasks.

VAS Questions	Easy < Hard Task	Easy = Hard Task	Easy > Hard Task
Imagine n (%)	55 (36.7)	69 (46.0)	44 (29.3)
4 Minutes n (%)	33 (22.0)	86 (57.3)	31 (20.7)
Difficulty n (%)	13 (8.6)	82 (54.7)	55 (36.7)

Visual analog scale (VAS) ratings from 0 to 100 in response to three questions asked at the end of each MI task scan: Imagine = How well were you able to imagine yourself doing the actual task?; 4 Minutes = Were you able to stay with the task the entire 4 min?; Difficulty = The activities you just visualized have a similar level of difficulty. In general, could you perform tasks of that difficulty?

## Data Availability

The data presented in this study are available upon request from the authors.

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
