# Peer review of "Effects of a Motor Imagery Task on Functional Brain Network Community Structure in Older Adults: Data from the Brain Networks and Mobility Function (B-NET) Study"

_brainsci, 2021, doi:10.3390/brainsci11010118_

Round 1

Reviewer 1 Report

The work presented has examined the neural correlates of task mobility using resting-state and motor-imagery task fMRI in157 healthy controls. Network connectivity and community structure analyses were conducted to characterize and compare responses corresponding to 3 different brain functional sub-networks of dorsal attention, default mode, and sensorimotor. Group findings showed decreased (global brain) network community structure during the motor-imagery task as compared to the resting state. Also, sensorimotor and default mode showed decreased consistency, whereas the attention network showed increased consistency during motor tasks as compare to resting-state task conditions. Findings support the use of fMRI connectivity analysis for mobility disability.

Findings are convincing, despite discrepancy with the initial hypothesis, and details are well explained. Overall, a well-written research paper. While I support this work for publication in the Journal of Brain Sciences, MDPI, I would like to bring 3 points to attention, and I hope the authors can address these at this point,

  • Fisher’s transform prior to second level (group) analysis seems to be required to convert correlation coefficients to normally distributed scores, to improve the normality of the data, and making subsequent statistics more robust (Whitfield-Gabrieli and Nieto-Castanon, 2012). I wonder this (or similar) technique was considered for the group analysis.
  • As for the fMRI task, I am not an expert but was not better to use a block design similar to previous works (as mentioned in the body document). This continuous design makes it a bit difficult to derive activities/networks of interests given that fMRI is poor with tracking temporal sequences. This has been pointed out in the discussions, maybe, it can be elaborated more as it is an important decision for similar studies in the future.
  • I wonder if relationships between performance and activities of the fMRI sub-networks were examined. For instance, was there any correlation between attention or/and motor networks with any measure corresponding to subjects’ performance?

Hope this helps.

Cite,

Whitfield-Gabrieli S, Nieto-Castanon A. 2012. Conn: a functional connectivity toolbox for correlated and anticorrelated brain networks. Brain Connect. 2:125–141.

Author Response

Thank you for taking the time to review our manuscript and provide excellent critiques. Please see the attached with our responses.

Reviewer 2 Report

The authors present analyses of multiple movement-related brain networks during a motor-imagery task using functional MRI and graph-theory based network analysis. Their focus is on a well-sized sample of the elderly. The manuscript is well written, and the findings will be of interests to the community. My primary complaint is that the authors may have missed a rare opportunity to take their analysis to the next level by exploring the relationship between the VAS scores and the brain network data on a trial-by-trial basis. In my opinion, this further analysis, that would require no new experiments, would improve the impact of the paper significantly.

Major criticisms

  1. Regions of interests were generated based on young adults while study participants were all 70+ years of age. If there are no prior studies available that would help map appropriate ROIs this use of younger subjects is understandable, but the authors should comment on the availability of such data in the age range the study focused on or on the potential similarities / differences between young adult and 70+ brain structure and the potential confounds this introduces in their work.
  2. The availability of self-reported ability to imagine the task or to follow it for the full 4 minutes provides a very interesting opportunity to test whether there is a relationship between the engagement of certain brain networks and the ability to complete the task on a trial-by-trial basis. It would greatly improve the paper if the authors provided insight into how brain network consistency (especially in DAN) relates to the VAS scores. This analysis may also give insight into why the DAN consistency appears lower during the hard task than the easy task (while the expectation might be the opposite).

Minor points

  1. The authors mention that global network statistics were assessed at three different thresholds, but they do not comment on whether any of the results or study outcomes were sensitive to these threshold differences.
  2. An alternative hypothesis as to why DAN consistency decreased during the harder task may be the involvement of a limbic / emotional component: if the participant could not complete such a task (eg. scale a flight of stairs without help) in years, imaging the action may result in a complex emotional response, disrupting the attentional network. If the authors have access to data pertaining potential changes in emotional state (functional data from limbic structures, heart rate, sweat etc.), this could enrich their analysis. Otherwise, this could be an interesting point to discuss.
  3. Line 178 typo: update

Author Response

(The authors gave the same response as above.)
